# An Ultrasensitive Room-Temperature H_2_ Sensor Based on a TiO_2_ Rutile–Anatase Homojunction

**DOI:** 10.3390/s24030978

**Published:** 2024-02-02

**Authors:** Xuefeng Wu, Ya Zhang, Menghan Zhang, Jianhu Liang, Yuwen Bao, Xiaohong Xia, Kevin Homewood, Manon Lourenco, Yun Gao

**Affiliations:** Ministry-of-Education Key Laboratory for the Green Preparation and Application of Functional Materials, Collaborative Innovation Center for Advanced Organic Chemical Materials Co-Constructed by the Province and Ministry, School of Materials Science & Engineering, Hubei University, Wuhan 430062, Chinaxhxia@hubu.edu.cn (X.X.); gaoyun@hubu.edu.cn (Y.G.)

**Keywords:** TiO_2_, homojunction, porous structure, charge transfer, H_2_ sensor

## Abstract

Metal oxide semiconductor hetero- and homojunctions are commonly constructed to improve the performance of hydrogen sensors at room temperature. In this study, a simple two-step hydrothermal method was employed to prepare TiO_2_ films with homojunctions of rutile and anatase phases (denoted as TiO_2_-R/A). Then, the microstructure of anatase-phase TiO_2_ was altered by controlling the amount of hydrochloric acid to realize a more favorable porous structure for charge transport and a larger surface area for contact with H_2_. The sensor used a Pt interdigital electrode. At an optimal HCl dosage (25 mL), anatase-phase TiO_2_ uniformly covered rutile-phase TiO_2_ nanorods, resulting in a greater response to H_2_ at 2500 ppm compared with that of a rutile TiO_2_ nanorod sensor by a factor of 1153. The response time was 21 s, mainly because the homojunction formed by the TiO_2_ rutile and anatase phases increased the synergistic effect of the charge transfer and potential barrier between the two phases, resulting in the formation of more superoxide (O_2_^−^) free radicals on the surface. Furthermore, the porous structure increased the surface area for H_2_ adsorption. The TiO_2_-R/A-based sensor exhibited high selectivity, long-term stability, and a fast response. This study provides new insights into the design of commercially competitive hydrogen sensors.

## 1. Introduction

As a renewable source of energy, hydrogen is widely used in chemical and metallurgical industries owing to its relatively clean and pollution-free nature and its high calorific value. It can also be stored on a large scale [1,2,3,4]. However, gaseous hydrogen is prone to leaking and is extremely flammable when it contacts oxygen in the atmosphere. Moreover, its tendency to explode upon encountering open flame poses a grave threat to human life and infrastructure. Therefore, detecting hydrogen gas in real-time at low concentrations is crucial to ensure the safe usage of hydrogen as an energy source [5,6].

Among the various gas detection methods, gas chromatographs use chromatographic columns to separate the individual gas components in a mixture to identify each component. Mass spectrometers identify gas molecules on the basis of their characteristic deflections from a magnetic field. Optical sensors utilize the change of optical properties of certain materials when they interact with H_2_ to detect H_2_. These methods typically require complex instrumentation and relatively large, expensive, and high maintenance [7,8,9], and electrochemical sensors generally have poor selectivity and long-term stability and are subject to interference from reducing gases [10]. Therefore, hydrogen sensors based on chemical resistance have been widely explored owing to their low cost, good stability, and excellent application prospects [11]. Among the numerous sensing materials, metal oxide semiconductors (MOSs) (e.g., TiO_2_, SnO_2_, and ZnO) are popular owing to their low cost and good sensitivity [12,13]. Among these, TiO_2_, a typical n-type semiconductor, is non-toxic, harmless, and inexpensive. Moreover, its unique performance in gas sensing is well-documented [14,15]. However, the response of TiO_2_ to hydrogen is limited by the rapid recombination of electron–hole pairs. Therefore, to improve the response of TiO_2_ to hydrogen, it is generally necessary to regulate its band structure and suppress the recombination rate of electrons and holes to ensure that more electrons combine with O_2_ to generate O_2_^−^. One effective method to suppress the recombination rate is to construct heterojunctions of TiO_2_ such as TiO_2_/MoS_2_ [16], Bi_2_MoO_6_/TiO_2_ [17], TiO_2_/Cu_2_O [18], and TiO_2_/SrTiO_3_ [19]. Owing to the advantages of the two types of metal oxide semiconductors, heterojunction composites exhibit significantly different physical and chemical properties and sensitivity to gases than their corresponding single-component counterparts. However, heterojunction composites contain many defects, which are not conducive to the performance of the heterojunction in terms of gas sensitivity [20,21].

A homojunction is formed between different crystal phases of the same semiconductor with different band structures. When carriers are transferred across a homojunction, their recombination rate would be much smaller than that at the heterojunction interface [22,23]. TiO_2_ has two stable phases, viz., rutile and anatase, with bandgaps of approximately 3.0 and 3.2 eV, respectively [21,24,25]. Therefore, in this study, we prepared composite materials with rutile–anatase TiO_2_ homojunctions for gas sensing. Biphasic TiO_2_ with homojunctions has a uniform composition and near-perfect lattice matching, which can ensure a reduced contact barrier, the regulation of the band structure, and effective charge transfer at the interface of the two phases (homojunction) [26,27].

Table 1 compares the performance of some previously reported MOS H_2_ sensors. It is seen that most MOS H_2_ sensors need to operate at high temperatures (>150 °C) [28,29,30,31,32,33]. Some reports indicate that modifying MOS sensors with precious metals can lower the operating temperature to room temperature [30]. However, there are few reports on MOS sensors that have a high response, a low detection limit, and a fast response time and operate at room temperature simultaneously. In this work, we designed and developed a growth-oriented material with rutile–anatase TiO_2_ homojunctions (denoted as TiO_2_-R/A) using a simple two-step hydrothermal method for H_2_ detection. In the second step of the synthesis, we controlled the amount of hydrochloric acid (HCl) to adjust the morphology of anatase-phase TiO_2_, promote charge transfer across the junction [34], and increase the contact area of the material surface with H_2_. Our results showed that the response of the bilayered mixed-phase TiO_2_ films with homojunctions to picomolar hydrogen was much greater than that of single-layer rutile-phase TiO_2_. The best performance in H_2_ sensing was achieved when anatase TiO_2_ was uniformly formed on the surface of the rutile TiO_2_ nanorods, at a hydrogen concentration of 2500 ppm, the response reached 1661. The characterization of the mixed-phase TiO_2_ material using X-ray diffraction (XRD), Raman spectroscopy, scanning electron microscopy (SEM), potentiometry, and X-ray photoelectron spectroscopy (XPS) is indicated, and the sensing mechanism is analyzed.

## 2. Materials and Methods

### 2.1. Fabrication of Sensing Layers

#### 2.1.1. Preparation of Rutile-Phase TiO_2_ Nanorod Arrays

First, a self-assembled rutile TiO_2_ nanorod array was grown on an FTO substrate, as reported previously [35]. Briefly, 28 mL of deionized water, 2 mL of ethanol, 30 mL of hydrochloric acid, and 1 mL of tetrabutyl titanate (TBOT) were mixed in a beaker and stirred to obtain a homogeneous precursor solution. Then, the precursor solution and FTO substrates, which were cleaned sequentially with acetone, ethanol, and deionized water, were placed in a 200 mL reactor and reacted at 150 °C for 8 h. After the reaction, the reactor was cooled to room temperature, and the FTO substrate was soaked in deionized water for 3 h. The rutile-phase TiO_2_ nanorod array was annealed in a 400 °C tube furnace for 20 min, and the prepared sample is referred to as NR-TiO_2_.

#### 2.1.2. Preparation of Rutile- and Anatase-Phase TiO_2_ Homojunctions

A known volume of hydrochloric acid (X = 10, 15, 20, 25, or 30 mL), 1 mL of concentrated sulfuric acid, and 1 mL of TBOT were added to a solution of ethanol (2 mL) and deionized water (28 mL) in a beaker and mixed evenly by stirring. Then, the NR-TiO_2_ sample was placed in a 200 mL reactor and reacted with the acidified TBOT solution at 150 °C for 8 h. After the reaction, the sample was rinsed several times with deionized water and then soaked in deionized water for 3 h. Finally, the sample was annealed in a tube furnace at 400 °C for 20 min. The samples thus prepared were denoted as TiO_2_(R/A-X), where R and A represent rutile and anatase phases and X represents the volume of HCl (10, 15, 20, 25, or 30 mL).

### 2.2. Characterization

XRD (D8-Advance, Bruker, Cu *Kα* line as the X-ray source), field-emission SEM (Sigma 500, Zeiss, Oberkochen, Germany), Raman spectrometry (CLY19 Lab RAM HR Evolution), and XPS were performed to characterize the microstructure, morphology, and chemical composition of the samples.

Platinum interdigitated electrodes were deposited onto thin films of TiO_2_(R/A-X) through DC magnetron sputtering. The standard test gas, 75% of H_2_ mixed with 25% of Ar or 0.75% of NH_3_, SO_2_, NO, C_2_H_6_, or NO_2_ dry gas mixed with 99.25% of Ar (Wuhan Xiangyun Industry and Trade Co., Ltd., Wuhan, China) was introduced into a 15 L sealed testing chamber via a computer-controlled mass flow controller (D07-7C, Beijing Seven Star Huachuang Flow Co., Ltd., Beijing, China). The sensing evaluation of the target gas was conducted at ambient temperature (25 °C) and environmental humidity (44%) unless specified otherwise. A Keithley 2400 multimeter was used to measure the fluctuation in sensor resistance while maintaining a constant voltage of 1 V across the sensor. Equation (1) can be used to ascertain the response of the sensor to H_2_ (S).
*S* = *R*_air_/*R*_gas_,(1)

In Formulas (1) and (2), the initial resistance of the sensor when exposed to air is denoted as *R*_air_, while the initial resistance in a H_2_ atmosphere is denoted as *R*_gas_. Δ*R* represents the recorded difference in resistance between the sensor in air and in an H_2_ atmosphere. The response time refers to the duration required for the sensor’s resistance to a decrease by 90% of Δ*R* from *R*_air_, whereas recovery time refers to the duration required for the sensor’s resistance to increase by 90% of Δ*R* from *R*_gas_.
∆*R* = *R*_air_ − *R*_gas_(2)

## 3. Results and Discussion

### 3.1. Characterization of the TiO_2_(R/A-X) Samples

Figure 1a presents the XRD patterns of the FTO substrate and the NR-TiO_2_ and TiO_2_(R/A-X) samples grown on the FTO substrates. The main sample peaks are consistent with those of rutile and anatase TiO_2_ (JCPDS No. 21-1276 and No. 21-1272). The diffraction peaks observed at 2*θ* = 36.1° and 62.7° correspond to the (101) and (002) crystallographic planes of rutile TiO_2_ (JCPDS 21-1276), and the peak at 2*θ* = 37.82° occurred due to the (004) plane of the anatase phase. A comparison with the standard patterns showed that the strongest peaks of the rutile and anatase phases did not appear in the XRD patterns of the prepared materials. The anatase (004) peak replaced the (101) peak as the strongest diffraction peak of the film. This indicates a tendency for the preferential growth of the nanorod in the (001) direction. The peak of 2*θ* = 62.7° in Figure 1a corresponds to the (002) crystal plane of rutile TiO_2_, which is consistent with previously reported results for rutile nanorod arrays, which tend to grow in the direction of (001) [35]. Therefore, the nanorods with rutile and anatase phases in the synthesized TiO_2_(A/R-X) thin films tended to grow in the (001) direction. As the main peak of the FTO substrate (PDF# 46-1088) at 2*θ* = 37.76° is very close to the (004) peak of the anatase phase, this region of Figure 1a was magnified. The locally magnified pattern revealed that the peaks observed near 2*θ* = 37.76° for the NR-TiO_2_ sample arose owing to the underlying FTO substrate. A comparison of the XRD patterns revealed that the diffraction peak of rutile (101) became progressively weaker with a gradual reduction in the amount of hydrochloric acid used in the second step of the synthesis. Further, the (004) diffraction peak of the anatase phase gradually increased, which was accompanied by a proportional decrease in the peak intensity of the rutile phase. This occurred because the inhibitory effect of Cl^−^ on anatase-phase TiO_2_ was weakened with the decrease in the amount of hydrochloric acid, which is consistent with previous reports [36,37]. The above phenomena were further confirmed by Raman spectroscopy.

Figure 1b shows the Raman spectra of the TiO_2_(R/A-X) samples. This is consistent with previous reports [38], which show a Raman band at 144 cm^−1^ corresponding to the *E*_g_ vibration mode of the anatase phase. The peaks at 448 and 608 cm^−1^ correspond to the *E*_g_ and A_1g_ vibrational modes of the rutile phase of TiO_2_. A comparison of the spectra revealed that as the amount of hydrochloric acid used in the second step increased, the main peak of the anatase-phase TiO_2_ (144 cm^−1^) increased gradually. Notably, the Raman spectrum of the TiO_2_(R/A-30 mL) sample did not show the anatase peak at 144 cm^−1^ because the content of the anatase-phase TiO_2_ in this sample was too low to be detected. Thus, the XRD and Raman spectroscopy results confirm the successful preparation of gas-sensitive films composed of anatase-phase TiO_2_ supported on the rutile-phase TiO_2_ nanorod array using a two-step hydrothermal method.

Figure 2 shows the surface morphologies of the NR-TiO_2_ and TiO_2_(R/A-X) samples. The surface of the NR-TiO_2_ sample shows the cauliflower-like morphology of the nanorod tips (Figure 2a) with diameters of approximately 100–200 nm. In this study, the formation of anatase-phase TiO_2_ on NR-TiO_2_ was controlled by varying the ratio of Cl^−^ to SO_4_^2−^. Figure 2 shows that as the amount of hydrochloric acid was decreased, a larger amount of anatase-phase TiO_2_ was generated in the reaction. When the amount of hydrochloric acid was decreased from X = 30 mL to X = 25 mL, the gaps between the nanorods became smaller. When the amount of hydrochloric acid was decreased further, below X = 25 mL, the gaps between the nanorods were gradually filled with anatase-phase TiO_2_.

Figure 3 shows the cross-sectional morphologies of the NR-TiO_2_ and TiO_2_(R/A-X) samples. The cross-sectional SEM image of the NR-TiO_2_ sample exhibited smooth nanorods (Figure 3a) with a height of approximately 2.3 μm. Figure 3b–f show the cross-sectional images of the TiO_2_(R/A-X) samples, which reveal that the morphology of the nanorods changed owing to modification with anatase-phase TiO_2_. As the amount of hydrochloric acid was decreased from X = 30 mL to X = 25 mL, the spaces between the nanorods gradually decreased. When X = 20 mL, anatase-phase TiO_2_ was enriched at the top of the nanorod, resulting in a distinct double-layer structure. With the reduction in the amount of hydrochloric acid, the anatase-phase-TiO_2_-enriched layer gradually became thicker, exhibiting an increase in thickness from 1.1 to 2.63 μm, as shown in Figure 3d–f. To compare the effects of the hydrochloric acid dosage (X = 25 mL and X = 30 mL) on anatase-phase TiO_2_ more clearly, we compared the micromorphologies of the NR-TiO_2_, TiO_2_(R/A-25 mL), and TiO_2_(R/A-30 mL) samples at a higher magnification (50 kx) (Figure 3g–i). The surface of the nanorods became rough due to modification with anatase-phase TiO_2_, and the surface roughness of the nanorods gradually increased with the decrease in hydrochloric acid dosage. Thus, SEM analyses confirmed that hydrochloric acid could significantly inhibit the growth of anatase-phase TiO_2_, which is consistent with the XRD and Raman results shown in Figure 1.

### 3.2. Sensing Properties

The hydrogen-sensing curves of NR-TiO_2_ and TiO_2_(R/A-X) samples are shown in Figure 4a–f. All samples exhibited a consistent n-type response to hydrogen. Owing to the reducing nature of hydrogen, the resistance of the sensor decreased when hydrogen entered the ventilation chamber and increased to the initial value when hydrogen exited the ventilation chamber. As shown in Figure 4a, the response of NR-TiO_2_ to hydrogen at room temperature was relatively poor, and the response values at 2500 and 20,000 ppm were 1.44 and 2.7, respectively (Appendix A). As shown in Figure 4b–f, the resistance of the TiO_2_(R/A-X) sample first increased and then decreased with the reduction in the amount of hydrochloric acid (X) used in the second step of the synthesis. Notably, in the sample prepared with X = 30 mL, only a very small number of anatase particles adhered to the sides of the rutile nanorods because the excessive chloride ions inhibited the growth of the anatase phase. In this case, the homojunction area formed between the rutile and anatase phases was relatively small, and the gap between the nanorods was large, resulting in relatively low resistance.

The response values of the NR-TiO_2_ and TiO_2_(R/A-X) samples during hydrogen sensing are shown in Figure 5a. TiO_2_(R/A-X) samples with homojunctions exhibited significantly higher response values than NR-TiO_2_. In particular, the highest response (1661) of 2500 ppm hydrogen was observed for the TiO_2_(R/A-25 mL) sample prepared using 25 mL of HCl. Further, with a further increase in hydrogen concentration, the response value of this sensor continued to show an increasing trend. At 20,000 ppm hydrogen, the response value of the TiO_2_(R/A-25 mL) sensor reached 76,702, which is 28,408 times that of NR-TiO_2_. It is worth noting that for the sample prepared with X = 30 mL, the response to hydrogen was only slightly improved compared with that of NR-TiO_2_ because only a very small amount of anatase TiO_2_ was formed on the rutile nanorod surface. According to the results shown in Figure 5, when the dosage of hydrochloric acid did not exceed 25 mL, the TiO_2_(R/A-X) sample showed a highly sensitive response to hydrogen at room temperature. In order to further evaluate the detection limit (LOD) of TiO_2_(R/A-25 mL), the response curve at a low concentration of 25–125 ppm H_2_ was measured, and the LOD was theoretically evaluated using Equation (3) [39,40], as shown in the Figure 5b.
(3)$$\mathrm{LOD}=\frac{3R_{noise}}{L_{slope}}$$
where $R_{noise}$ is the measured noise of the sensor and $L_{slope}$ is the slope of the linear fitting curve. The $R_{noise}$ as 0.01 is calculated using the equation [39,40] below: (4)$$R_{noise}=\sqrt{\frac{\Sigma {\left(R_{i}-R\right)}^{2}}{N}}$$
where $R_{i}$ is the experimental data (i.e., various responses of H_2_ concentration) and R is the fitting value based on Figure 5b. Calculation reveals that the LOD of the TiO_2_(R/A-25 mL) sensor is ~6.3 ppm.

As shown in Figure 5c,d, the TiO_2_(R/A-X) samples exhibited a slightly longer response time and a longer recovery time than NR-TiO_2_ (see Appendix A). This was the case because the increased response causes the resistance to fluctuate over a wider range, and the resistance of the system takes more time to reach the equilibrium value, especially in environments with high hydrogen concentrations, where this phenomenon is much more evident. In summary, the response values of NR-TiO_2_ at all tested hydrogen concentrations ranged from 1 to 2, whereas the response values of the TiO_2_(R/A-X) samples at the same tested hydrogen concentrations were significantly better.

After the surface of the rutile TiO_2_ nanorods was coated with anatase TiO_2_, TiO_2_(R/A-25 mL) exhibited exceptional sensing properties. It is crucial to comprehend its key determinants. The sensing performance of this sample can be attributed to the presence of type II alternating band alignment of ~0.4 eV between anatase and rutile phases and the higher electron affinity of anatase-phase TiO_2_. Therefore, electron flow occurs from the rutile phase to the surface anatase phase, as shown in Figure 6a,b. Because the rutile and anatase phases of TiO_2_ have different band gaps (*E*_g_) and different valence band and conductive band energies (*E*_V_ and *E*_C_, respectively), when the rutile-phase TiO_2_ nanorod contacts the anatase-phase TiO_2_ layer, an internal potential is established at the junction of the two phases. Owing to the continuous flow of electrons from the rutile to the anatase phase of TiO_2_, an electron depletion layer is formed at the rutile phase TiO_2_ nanorods, whereas an electron accumulation layer is formed at the anatase layer, as illustrated in Figure 6b. 

To investigate this, XPS analyses were conducted on samples of NRs-TiO_2_, anatase phase TiO_2_, and TiO_2_(R/A-X), as depicted in Figure 6c,d. The O1s spectrum of the NRs-TiO_2_ sample (Figure 6c) exhibited two prominent peaks at 529.71 eV and 530.91 eV, corresponding to lattice oxygen (Ti-O-Ti) and hydroxide (Ti-OH), respectively. In the Ti2p spectrum (Figure 6d), two distinct nuclear level signals may be observed at 464.30 eV and 458.54 eV, which can be attributed to the Ti2p3/2 and Ti2p1/2 levels of Ti^4+^, respectively.

Consistent with previously reported findings [41], the O1s spectra of anatase-phase TiO_2_ exhibited higher binding energy peaks for lattice oxygen (Ti-O-Ti) and hydroxyl groups (Ti-OH). For both the sensor TiO_2_(R/A-10 mL) and anatase-phase TiO_2_ samples, no significant change in peak values was observed; this can be explained by the aforementioned microscopic morphology, in which a rutile-phase nanorod was present on the top surface covered with anatase-phase TiO_2_ with a thickness of approximately 2.6 μm. Consequently, the XPS analysis only detected signals from the anatase phase of TiO_2_. 

In addition, the formation of a homojunction between rutile phase TiO_2_ and anatase-phase TiO_2_ in the TiO_2_(R/A-25 mL) sample results in negative shifts in the binding energies of O*1s* (−0.27 eV), Ti2p3/2 (−0.22 eV), and Ti2p1/2 (−0.24 eV) compared to pure anatase-phase TiO_2_. This peak shift further confirms electron flow from rutile-phase TiO_2_ to anatase-phase TiO_2_.

To explain the response mechanism of the sensor to H_2_ more clearly, one usually needs to calculate the Schottky barrier formed at the two-phase contact surface. Therefore, we tested the I-V change curve of TiO_2_(R/A-X) at room temperature to estimate the difference in barrier height between in air (Figure 7a) and H_2_ (2500 ppm) atmosphere (Figure 7b). The *I*–*V* curves of the entire series of TiO_2_(R/A-X) samples changed significantly before and after exposure to H_2_ atmosphere. By observing the phenomenon of the nonlinear relationship between I-V, we can consider that there was a potential barrier [42]. The effective height of the barrier can be calculated by Equation (5) [43,44]. 

(5)$$\Phi B=\frac{kT}{q}\;\ln({\frac{A}{I}}_{s}A^{*}T^{2}),$$
where *k* represents the Boltzmann constant, *T* represents the test temperature, *A* represents the area where the diode is constructed, *I_s_* represents the saturation current, and *A** represents the Richardson constant of rutile-phase TiO_2_. In this series of sensors, the value of A was equal to 2.5 cm × 2.5 cm = 6.25 cm^2^, T at room temperature usually takes a value of 300 K, and *A** = (m*/m) A/cm^2^ K, for rutile-phase TiO_2_ m*/m = 20. The saturation current can be calculated by the following formulas (6) and (7): (6)$$I=I_{s}\;\exp(\frac{qV}{nkT}),$$
(7)$$\ln I=\ln I_{s}+\frac{qV}{nkT}$$

In the formula given above, q represents the amount of charge carried by the electron and n represents the ideal factor. There are two main factors that affect the height of the effective barrier in this series of sensors. First, H_2_ reacts with the topmost Pt electrode to form PtHx, which decreases the height of the Schottky barrier [44,45,46]. Second, the change in the charge concentration in the anatase-phase TiO_2_ layer also changes the barrier height. Therefore, the potential barrier of the TiO_2_(R/A-X) sample in air showed a trend of first increasing and then decreasing with increasing anatase TiO_2_ content. The highest potential barrier was observed for TiO_2_(R/A-25 mL). The increase in the barrier heights of TiO_2_(R/A-30 mL) and TiO_2_(R/A-25 mL) was due to the gradual increase in the generation of anatase TiO_2_ and the resultant gradual increase in the interface area between the rutile and anatase phases. As the generation of anatase TiO_2_ was increased further by reducing the amount of HCl, lateral current leakage occurred due to the accumulation of anatase TiO_2_ on the surface of the rutile-phase TiO_2_ nanorods, which decreased the barrier height.

Figure 7c,d show the calculated effective barrier heights of different samples in air and 2500 ppm H_2_ environments and the correlation between the barrier heights of the samples and the corresponding response of the sensor. Under 2500 ppm H_2_, the barrier heights of all samples were less than those in air, and the barrier difference was consistent with the curve of the sensing response. The largest change in the barrier height of 0.30 eV was observed for the TiO_2_(R/A-25 mL) sample, which exhibited the highest response to H_2_.

Figure 8a,b show the changes in the morphology of the anatase-phase TiO_2_ layer with the reduction in the amount of hydrochloric acid used in the synthesis. As the amount of hydrochloric acid was gradually decreased, anatase-phase TiO_2_ was first coated on the surface of rutile nanorods (Figure 8a), As the amount of hydrochloric acid continued to decrease, the deposition of anatase-phase TiO_2_ on the surface of the rutile nanorods gradually increased, resulting in a double-layer structure (Figure 8b). In the schematic shown in Figure 8a, anatase-phase TiO_2_ completely covers the surface of rutile-phase TiO_2_ nanorods to form a porous structure. In this situation, only the longitudinal current *I*_1_ along the nanorods is present (Figure 8a). As the amount of hydrochloric acid was decreased, anatase-phase TiO_2_ generated by the reaction was not only deposited on the nanorod surface but also filled the gaps between the nanorods. This structure not only has longitudinal current *I*_1_ along the nanorods but also generates transverse current *I*_2_ (Figure 8b). This model explains the initial increase followed by the decrease in the resistance of the TiO_2_(R/A-X) samples, as shown in Figure 4. The schematic in Figure 8c,d shows the response mechanism of the sensor to H_2_. Electrons flow from rutile-phase TiO_2_ nanorods to the surface anatase-phase TiO_2_ layer and react with O_2_ adsorbed to the surface to generate a large amount of O_2_^−^ species in air. At this stage, a large depletion layer is formed on the sides of the rutile-phase nanorods (Figure 8c), and the resistance of the system increases. When H_2_ enters the test chamber, H_2_ reacts with O_2_^−^ on the surface of the sensor so that the released electrons return to the rutile-phase TiO_2_ nanorods, the depletion layer becomes smaller, and the resistance of the system decreases (Figure 8d). The associated reaction can be represented as follows [44,45,46].
O_2_ + e^−^→O_2_^−^,(8)
2H_2_ + O_2_^−^→2H_2_O + e^−^.(9)

According to the micromorphology of the TiO_2_(R/A-25 mL) sample in Figure 2 and Figure 3, rutile-phase TiO_2_ nanorods are tightly wrapped by anatase-phase TiO_2_, which has a porous structure that is more conducive to contact with H_2_.

The selectivity of the TiO_2_(R/A-25 mL) sample was tested at 2500 ppm concentrations of H_2_, NO, SO_2_, C_2_H_6_, NH_3_, and NO_2_. The dynamic curve of the resistance is shown in Figure 9a, and the response values of the sensor corresponding to different gases are shown in Figure 9b. The sensor exhibited an N-type response to reducing gases, and TiO_2_, as a typical N-type semiconductor, exhibited this phenomenon in line with the oxygen adsorption theory [44]. However, NO_2_, an oxidizing gas, also exhibited an N-type response, which contradicts the theory of oxygen adsorption. This phenomenon was a result of the catalytic action of the Pt electrode. The highly oxidizing NO_2_ gas reacts preferentially with Pt, and NO_2_ is broken down into NOx and O, thereby transferring electrons to the surface of TiO_2_ and reducing the resistance of the film. When the sensor was exposed to NH_3_ and C_2_H_6_ atmospheres, the initial resistance of the sensor did not change, reflecting that the sensor was not responsive to these two gases. Because these two gas molecules show high stability at room temperature, they are less likely to break the chemical bond and cause a change in the resistance of the TiO_2_ [45]. As shown in Figure 9b, at the same concentration of 2500 ppm, the response of the TiO_2_(R/A-25 mL) sensor to hydrogen was 109 times greater than that of NO, 230 times more than that of SO_2_, and 503 times greater than that of NO_2_, indicating that the TiO_2_(R/A-25 mL) sensor has excellent selectivity to H_2_.

Figure 9c shows the results of the cyclic testing of the TiO_2_(R/A-25 mL) sample with 25, 250, 1250, 2500, and 5000 ppm of H_2_. The samples were exposed to room-temperature air for 170 d and tested in four cycles. The response value was then compared with the initial value to calculate the attenuation in the response (Figure 9d). As shown in Figure 9d, when the hydrogen concentrations were 25, 250, 1250, 2500, and 5000 ppm, the decay rates of the homojunction were 7, 8.5, 11, 11.4, and 20%, respectively, indicating that the homojunction is very stable at room temperature. Figure 9e shows the response of the TiO_2_(R/A-25 mL) sensor to H_2_ under different humidities of 44, 62, and 82%, respectively. As the relative humidity was increased gradually from 44 to 82%, the initial resistance of the sample decreased gradually. Figure 9f presents the response value of the sample under different humidities. As the humidity increased, the response value of the sensor to hydrogen decreased significantly because of the competitive adsorption of water molecules on the TiO_2_ film surface, which attenuates the H_2_-sensing performance. Due to the increase in humidity, water molecules were adsorbed on the surface of the sensor, resulting in reduced oxygen adsorption at the active site on the surface of the TiO_2_. As a result, the reaction between oxygen and H_2_ molecules weakened rapidly, and the drop in the surface resistance of TiO_2_ was reduced. Under conditions of 44 and 62% humidity, the sensor response was still very good, reflecting the good moisture resistance of the sample, which essentially meets the requirement of hydrogen detection in daily life.

## 4. Conclusions

A TiO_2_ rutile–anatase dual-phase homojunction material was prepared for hydrogen sensing. The key sensing layers with different TiO_2_ phases were prepared by the in situ modification of rutile TiO_2_ nanorod films produced by a hydrothermal method. In the second step of the hydrothermal reaction, the thickness and morphology of the second anatase phase were controlled by changing the volume of HCl used to control the growth of this phase. The relationship between the barrier height and microstructure of the TiO_2_ rutile–anatase dual-phase material was investigated through various measurements. The experimental results show that the interface of the TiO_2_ nanocrystalline rutile phase and anatase phase forms a uniform homojunction, and a porous structure more conducive to carrier transport is formed, thus enhancing the response of the sensor to hydrogen. The TiO_2_ rutile–anatase heterojunction plays a crucial role in regulating the barrier height under H_2_ conditions. Among the different samples, the TiO_2_(R/A-25 mL) sample exhibited the best response to 2500 ppm hydrogen (as high as 1661) with good long-term stability and selectivity. The results of this study provide novel insights to support the design of commercially competitive hydrogen sensors.

## Figures and Tables

**Figure 1 sensors-24-00978-f001:**
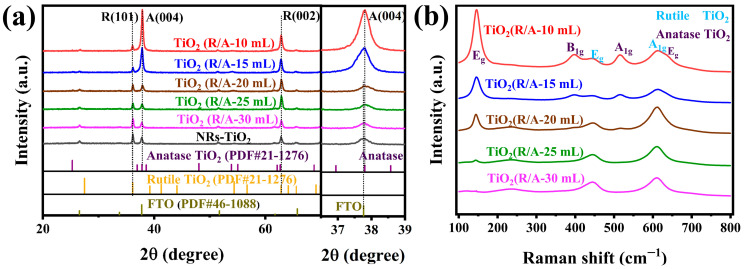
(**a**) XRD spectrum of the FTO substrate, NR-TiO_2_/FTO, and TiO_2_(R/A-X)/FTO. (**b**) Raman spectra of TiO_2_(R/A-X).

**Figure 2 sensors-24-00978-f002:**
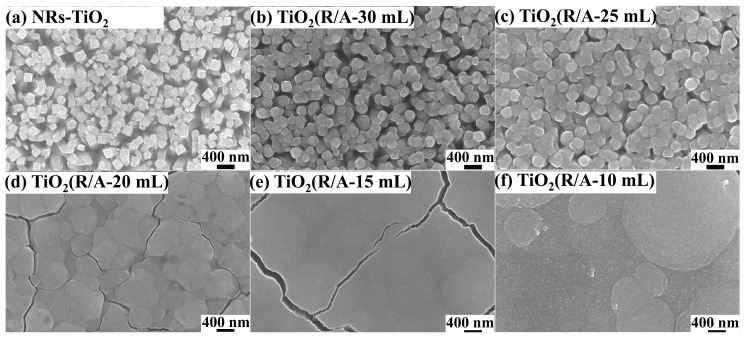
SEM images showing the surface morphologies of (**a**) NR-TiO_2_, (**b**) TiO_2_(R/A-10 mL), (**c**) TiO_2_(R/A-15 mL), (**d**) TiO_2_(R/A-20 mL), (**e**) TiO_2_(R/A-25 mL), and (**f**) TiO_2_(R/A-30 mL).

**Figure 3 sensors-24-00978-f003:**
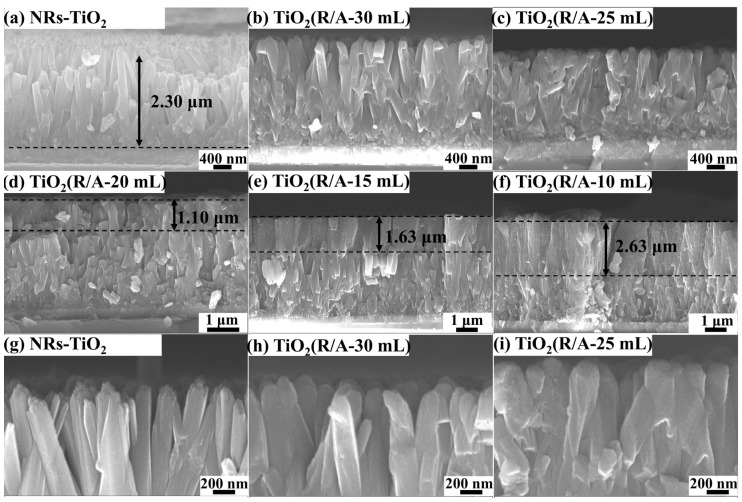
SEM images showing the cross-sectional morphologies of (**a**) NR-TiO_2_, (**b**) TiO_2_(R/A-30 mL), (**c**) TiO_2_(R/A-25 mL), (**d**) TiO_2_(R/A-20 mL), (**e**) TiO_2_(R/A-15 mL), (**f**) TiO_2_(R/A-10 mL), and the cross-sectional morphologies with higher magnification (50 kx) of (**g**) NR-TiO_2_, (**h**) TiO_2_(R/A-30 mL) and (**i**) TiO_2_(R/A-25 mL).

**Figure 4 sensors-24-00978-f004:**
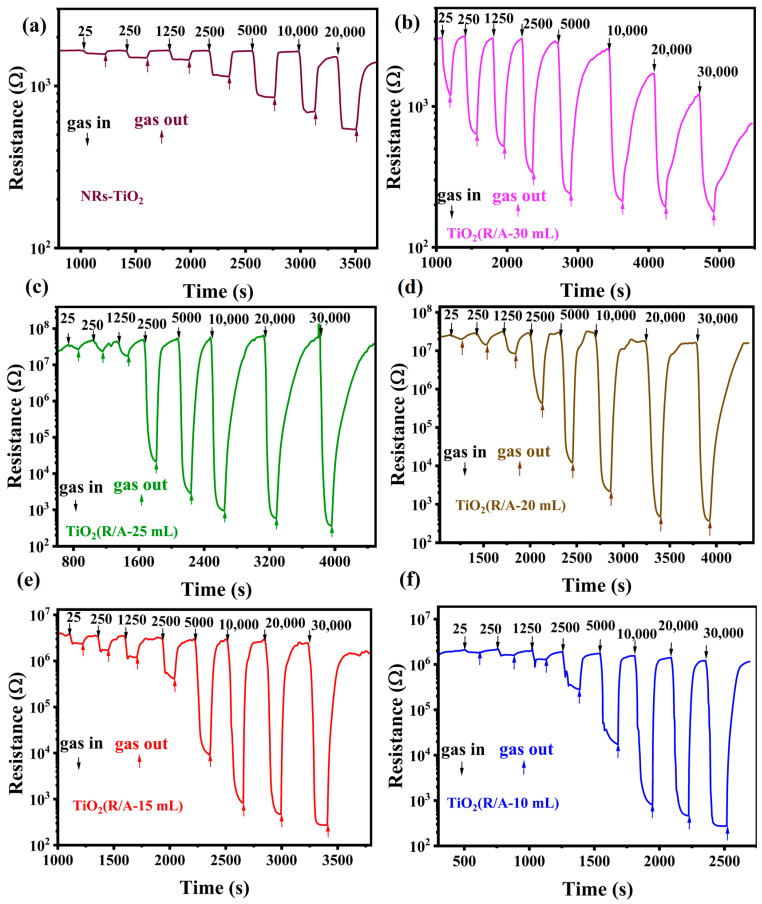
Hydrogen-sensing characteristics of the TiO_2_(R/A-X) sensors at 25 °C. Change in resistance with time at different hydrogen concentrations for (**a**) NR-TiO_2_, (**b**) TiO_2_(R/A-30 mL), (**c**) TiO_2_(R/A-25 mL), (**d**) TiO_2_(R/A-20 mL), (**e**) TiO_2_(R/A-15 mL), and (**f**) TiO_2_(R/A-10 mL).

**Figure 5 sensors-24-00978-f005:**
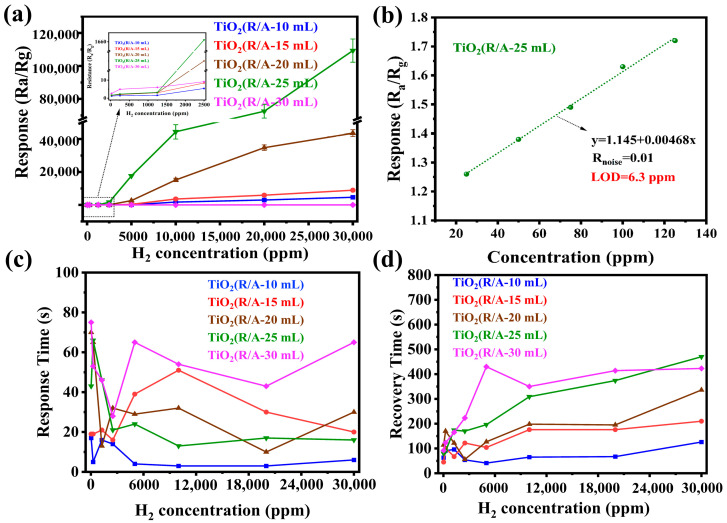
Hydrogen-sensing performances of the TiO_2_(R/A-X) sensors. (**a**) Response values at different hydrogen concentration from 25 to 30,000 ppm and (**b**) LOD of TiO_2_(R/A-25 mL) at low H_2_ concentration. (**c**,**d**) Response and recovery times of the sensors at different hydrogen concentration from 25 to 30,000 ppm.

**Figure 6 sensors-24-00978-f006:**
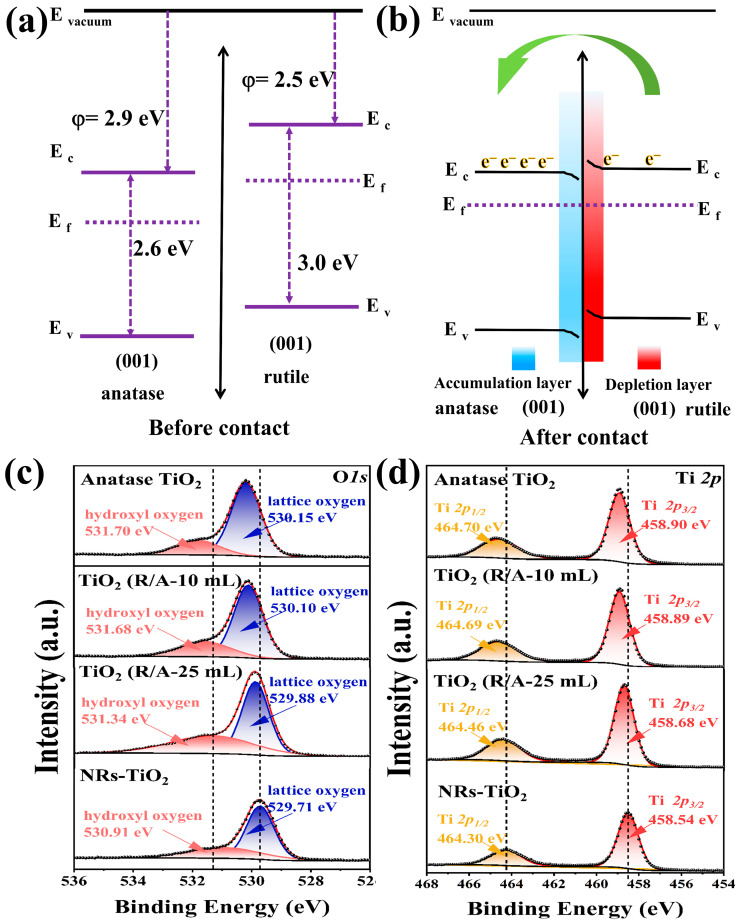
(**a**,**b**) Bandgaps (*E*_g_) and the energies of the valence band (*E*_v_) and conduction band (*E*_c_) for the rutile and anatase phases of titania. (**c**) O1s and (**d**) Ti2p XPS profiles of the NR-TiO_2_ and TiO_2_(R/A-25 mL) samples.

**Figure 7 sensors-24-00978-f007:**
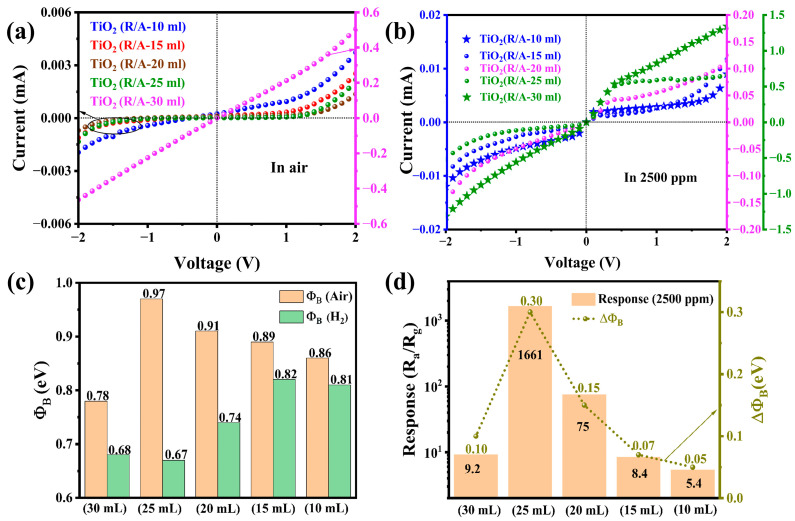
(**a**,**b**) Representation of I-V characteristics of the TiO_2_(R/A-X) sensor in air and H_2_ (2500 ppm), respectively. (**c**) Changes in the Schottky barrier height of the TiO_2_(R/A-X) sensor before and after exposure to H_2_ at 2500 ppm. (**d**) Relationship between the barrier height of the TiO_2_(R/A-X) sensor and response value.

**Figure 8 sensors-24-00978-f008:**
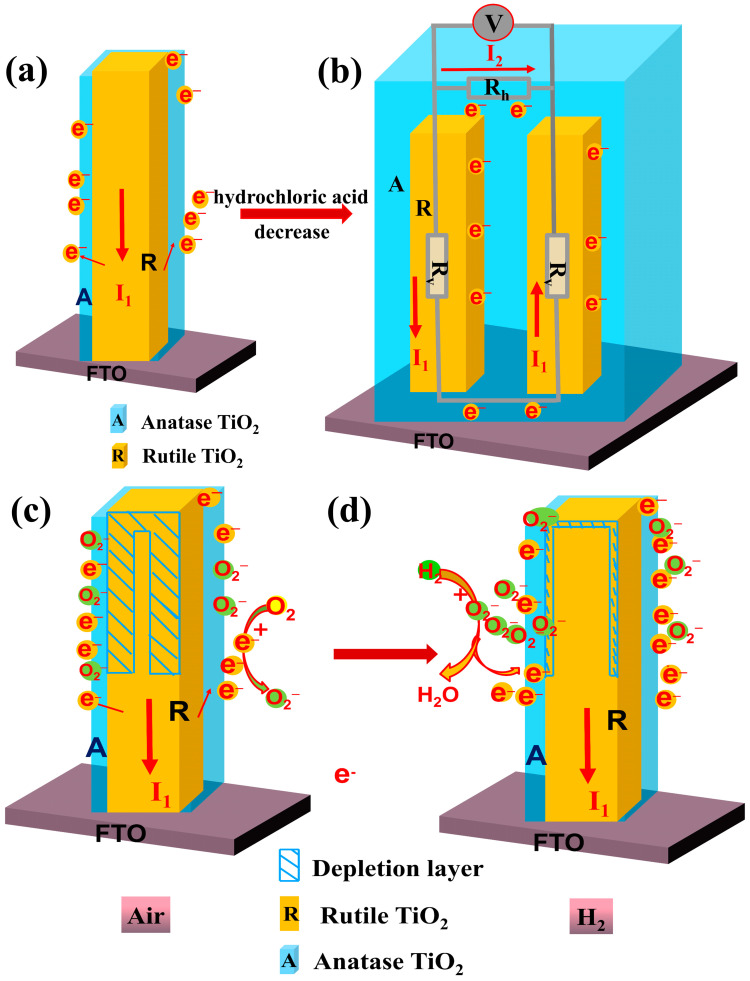
(**a**,**b**) Schematic showing the change in the morphology of TiO_2_ nanorods during the second step of the reaction. (**c**,**d**) Schematic illustration of the mechanism of the sensor response.

**Figure 9 sensors-24-00978-f009:**
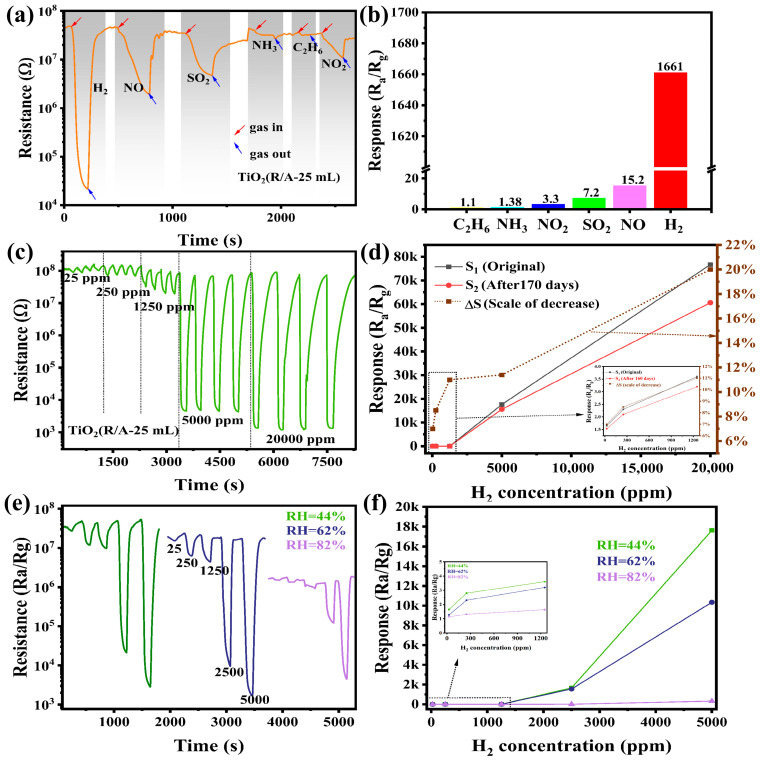
Sensing performance of TiO_2_(R/A-25 mL) for H_2_, NO, SO_2_, NH_3_, C_2_H_6_, and NO_2_ at 2500 ppm: (**a**) resistance change with time and (**b**) sensor response to different gases. (**c**) Cycling stability of TiO_2_(R/A-25 mL) at 25, 250, 1250, 2500, and 5000 ppm hydrogen after being placed in air for 170 d. (**d**) Comparison of the sensor response after 170 d with the initial response and percentage decline in the response. (**e**) Resistance changes of the TiO_2_(R/A-25 mL) sensor in response to H_2_ under different humidities and (**f**) responses of the sample under different humidities.

**Table 1 sensors-24-00978-t001:** Comparison of sensing performances of H_2_ sensors.

Materials	T (°C)	Concentration (ppm)	Response (R_a_/R_g_)	Ref.
ZnO-SnO_2_	150	10,000	52	[28]
SnO_2_-Co_3_O_4_	500	350	4.5	[29]
Pt@NiO	RT	5000	4.25	[30]
1 at.% Pt/SnO_2_	350	100	56.5	[31]
1.0 at% Pd/SnS_2_/SnO_2_	300	500	95	[32]
3 at% PdO/WO_3_	150	50	77	[33]
TiO_2_(R/A-25 mL)	RT	2500	1661	This work

## Data Availability

The data presented in this study are available upon request from the corresponding author. The data are not publicly available due to privacy and ethical restrictions.

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
