# Peer review of "An Ultrasensitive Room-Temperature H2 Sensor Based on a TiO2 Rutile–Anatase Homojunction"

_sensors, 2024, doi:10.3390/s24030978_

Round 1

Reviewer 1 Report

Comments and Suggestions for Authors

The  introduction must be improved. It is necessary to introduce existing hydrogen detection methods, including but not limited to sensors. What improvements have been made to this work  compared to existing methods?

The quality and clarity of ALL the images need to be improved. Texts in the images  are not clear. Multiple measurements should be taken and error bars should be drawn in some of the graphs.

A curve should include at least five points. And each curve in Figures 9d and 9f has only three points. This work only tested three hydrogen concentrations, which is not enough to determine the measurement performance of the sensor

This article does not provide data on the lower detection limit and linear range. These data ( detection limit, linear range, sensitivity) is important to evaluate a sensor.

The detection of hydrogen by this sensor requires the use of oxygen in the environment (line 320-321) . Therefore, the effects of oxygen concentration and interference of  other oxidizing gases in the environment need to be discussed.

What is the temperature range for the operation of this sensor?

Comments on the Quality of English Language

The  introduction must be improved. It is necessary to introduce existing hydrogen detection methods, including but not limited to sensors. What improvements have been made to this work  compared to existing methods?

The quality and clarity of ALL the images need to be improved. Texts in the images  are not clear. Multiple measurements should be taken and error bars should be drawn in some of the graphs.

A curve should include at least five points. And each curve in Figures 9d and 9f has only three points. This work only tested three hydrogen concentrations, which is not enough to determine the measurement performance of the sensor

This article does not provide data on the lower detection limit and linear range. These data ( detection limit, linear range, sensitivity) is important to evaluate a sensor.

The detection of hydrogen by this sensor requires the use of oxygen in the environment (line 320-321) . Therefore, the effects of oxygen concentration and interference of  other oxidizing gases in the environment need to be discussed.

What is the temperature range for the operation of this sensor?

Reviewer 2 Report

Comments and Suggestions for Authors

In this study, a simple hydrothermal method was employed to prepare TiO2 films with homojunctions of rutile and anatase phases. It shows the high performance for H2 sensing, The response time was 21 s. The enhanced mechanism is mainly because the homojunction formed by the TiO2 rutile and anatase phases increased the synergistic effect of charge transfer and potential barrier between the two phases, resulting in the formation of more superoxide free radicals on the surface. I recommend its publication on Sensors, after the authors figured out the following issues.

(1) The Figure resolution is rather low.

(2) Is there any difference in the R-TiO2 and A-TiO2, and will the oxygen defects be formed in the composite?

(3) Why the sensor response vs concentration donot show the linear relationship?

(4) Will the humidity influence the sensing response for the TiO2 sensors.

(5) In the bar plot, the NO2 sensing response should not be indicated with Ra/Rg, because it is a oxidizing gas not like H2.

Comments on the Quality of English Language

NO

Reviewer 3 Report

Comments and Suggestions for Authors

The paper makes a good impression. The sensory responses are large enough for room temperature, and the recovery time is quite low. The material is new and it is well characterized.

It is not entirely clear why hydrochloric acid was chosen for treatment the material. It is known that chloride anions remain in the material, they have high electrical conductivity and reduce the signal of the semiconductor. In this regard, precursors that do not contain chloride anions, sodium cations, and so on are traditionally used in the synthesis of gas-sensitive materials. It is possible that the use of hydrochloric acid is associated with the use of a gas-sensitive material at room temperature, and yet the choice of nitric acid seems preferable because it decays quite easily to form volatile components.

Round 2

Reviewer 1 Report

Comments and Suggestions for Authors

 this edition is acceptable

Comments on the Quality of English Language

 this edition is acceptable